# API-Assisted Code Generation for Question Answering on Varied Table Structures

**Yihan Cao**[*]    **Shuyi Chen**[*]    **Ryan Liu**[*]
**Zhiruo Wang**    **Daniel Fried**
Carnegie Mellon University
{yihanc,shuyic,ryanliu,zhiruow,dfried}@andrew.cmu.edu

## Abstract

A persistent challenge to table question answering (TableQA) by generating executable programs has been adapting to varied table structures, typically requiring domain-specific logical forms. In response, this paper introduces a unified TableQA framework that: (1) provides a unified representation for structured tables as multi-index Pandas data frames, (2) uses Python as a powerful querying language, and (3) uses few-shot prompting to translate NL questions into Python programs, which are executable on Pandas data frames. Furthermore, to answer complex relational questions with extended program functionality and external knowledge, our framework allows customized APIs that Python programs can call. We experiment with four TableQA datasets that involve tables of different structures — relational, multi-table, and hierarchical matrix shapes — and achieve prominent improvements over past state-of-the-art systems. In ablation studies, we (1) show benefits from our multi-index representation and APIs over baselines that use only an LLM, and (2) demonstrate that our approach is modular and can incorporate additional APIs.

## 1 Introduction

Tables are an important and widely used format for storing and retrieving information. However, since they are often constructed to present information in a visually effective way, they consequently occur in diverse formats (Chen and Cafarella, 2013; Nishida et al., 2017; Wang et al., 2021b). Thus, to effectively answer questions about all tabular information, we must consider information stored in relational (Pasupat and Liang, 2015), matrix, and hierarchically indexed tables (Cheng et al., 2022), and also address scenarios where multiple tables are presented conjointly (Yu et al., 2018).

In the past, works have focused on achieving strong results on datasets with particular table struc-

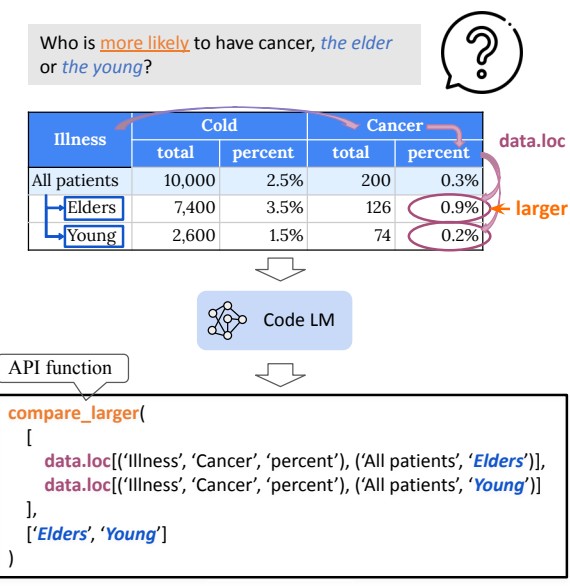

Figure 1: Our approach answers questions about complex tables by representing the tables as a Pandas multi-index data frame, and using a code generation LM to translate the question into a Python program that uses assistant API functions to operate on the data frame.

tures and required tailoring logical forms to each specific type of table structure (Wang et al., 2015; Guo et al., 2019). These methods struggle to work on tables with structures outside of their original domain. For example, the neural-symbolic machine (NSM) approach designed for relational tables (Liang et al., 2017) only achieves 29.2% accuracy on the hierarchical matrix table dataset HiTab (Cheng et al., 2022) due to the ineffectiveness of its logical forms on hierarchical tables.

Our work focuses on developing a unified framework to solve TableQA tasks across diverse table structures using Python as an intermediate language. In contrast with previous approaches that use custom logical forms to query table data (Guo et al., 2019; Cheng et al., 2022), we propose to query entries and perform operations within the

---

[*]Equal contribution.

widely used Python Pandas library. This allows us to leverage the strong few-shot Python generation capabilities of large code generation models, whilst saving costs by using only a few training examples. An overview of the framework is presented in Figure 1, with more details in Figure 3. Our framework consists of three main parts:

First, we transform tables with varied structures into a unified multi-index data frame representation adopted from the Python Pandas library (§3.1). The multi-index objects can effectively retain the structural information in a wide range of table formats. From the implicit first step illustration in Figure 1 and more detailed one in Figure 2c, we convert tables from a hierarchical format to the multi-index representation, which will enable generated code to successfully query elements in the table.

Second, we translate TableQA questions into Python programs as an executable intermediate language by prompting code generation models (§3.2). Specifically, we use a few-shot paradigm where we provide the multi-index headers of the table, a few rows of the table in textual form, the question, and exemplar program annotations. As shown in Figure 1, the table and question are input to the code generation model along with the few-shot prompt, and the generated Python code uses the Pandas `data.loc` function to query the appropriate cells.

Third, we introduce assistant API functions to extend the capabilities of our framework beyond Python Pandas and achieve broader coverage over various TableQA tasks. These functions enable a model-generated program to query external knowledge and perform various additional operations on the multi-index representation. In this paper, we demonstrate the usage of two simple types of API functions, Operation APIs and QA API (§3.3), and show through ablations that they increase the performance of our framework across various datasets. For example in Figure 1, the model outputs code using one of our API functions, `compare_larger`, to solve the question.

We evaluate our method across relational, hierarchical, and matrix tables, as well as multiple table paradigms using the WikiTableQuestions (WikiTQ) (Pasupat and Liang, 2015), HiTab (Cheng et al., 2022), AIT-QA (Katsis et al., 2022), and Spider (Yu et al., 2018) datasets (§4). For code generation models, we use a more capable proprietary model, CODEX (Chen et al., 2021), and an open-source model, STARCODER (Li et al., 2023).

We find that our framework surpasses the state-of-the-art few-shot baselines on the HiTab, AIT, and Spider datasets, achieving absolute improvements of 24.5%, 26.2%, and 2.3%, respectively, while retaining non-trivial performance on WikiTQ. Furthermore, we perform an ablation study on the API functions we introduced and find that they bring significant improvements across datasets and models. Our framework also allows the use of existing API operations within Pandas in a modular way. For example, we show that combining Pandas' SQL interface API with our multi-index representation and APIs improves performance by 3.7–7.6% on relational datasets.

## 2 Problem Setting and Datasets

In this section, we provide a formal description of the TableQA tasks (§2.1) and describe the four datasets used in our experiments, containing tables of varied structures (§2.2).

### 2.1 Problem Statement

Each example in a TableQA task contains a table $t$ and a question $q$ about the table, and aims to generate the correct answer $a$ to the question. While both $q$ and $a$ are textual sequences, the table $t$ can vary in structure — from relational tables (Figure 2a, Figure 2b) to hierarchical matrix tables (Figure 2c).

The common *symbolic* approach to TableQA generates an intermediate program $p$ from a model $M$ using the table and question, $p = M(q, t)$. The program is then executed on $t$ to yield the answer $a' = exec(p, t)$. In traditional methods, tables are serialized to text sequences by cell string concatenation, and programs are logical forms specifically designed for certain domains. To preserve and leverage more structural information, we represent tables as Pandas multi-index objects, $t_m$, and generate Python programs accordingly.

### 2.2 QA over Varied Table Structures

We use four diverse TableQA datasets to evaluate our approach: WikiTQ (Pasupat and Liang, 2015), HiTab (Cheng et al., 2022), AIT-QA (Katsis et al., 2022), and Spider (Yu et al., 2018). As listed in Table 1, these datasets cover a broad spectrum of table structures and topical domains, thereby presenting various challenges associated with different table structures.

**WikiTableQuestions (WikiTQ)** WikiTQ (Pasupat and Liang, 2015) contains relational tables with

| Dataset | Size | | Table Info | |
|---|---|---|---|---|
| | dev | test | structure | source |
| WikiTQ (Pasupat and Liang, 2015) | 2,381 | 4,344 | relational, single | Wikipedia |
| HiTab (Cheng et al., 2022) | 1,671 | 1,584 | hierarchical matrix | Stat. reports, Wikipedia |
| AIT-QA (Katsis et al., 2022) | — | 515 | hierarchical | Airline documents |
| Spider (Yu et al., 2018) | 1,034 | — | relational, multiple | College data, WikiSQL |

\* Spider has an undisclosed online test set.
\*\* We treat the entire AIT-QA dataset as a single test set in our evaluations.

Table 1: Statistics and properties of the datasets we use to evaluate our approach, encompassing varied table structures and sources.

relatively simple structures. All tables are collected from Wikipedia articles. The questions usually involve basic arithmetic operations, such as sum and max, and sometimes require compositional reasoning over table entries. Since our method does not require any training, we use the standard validation set and test set with 2,381 and 4,344 samples.

**HiTab** (Cheng et al., 2022) involves hierarchical matrix tables collected from statistical reports and Wikipedia articles, covering diverse domains such as economics, education, health, science, and more. HiTab challenges existing methods with its broad set of operations, complex table structures, and diverse domain coverage. It contains 3,597 hierarchical tables with 10,672 questions. We evaluate on their test set containing 1,584 samples.

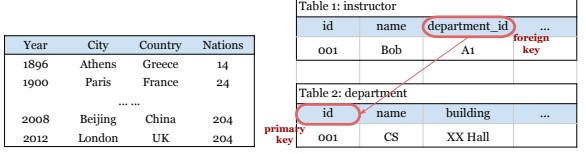

(a) A relational table.    (b) A multi-table relation.

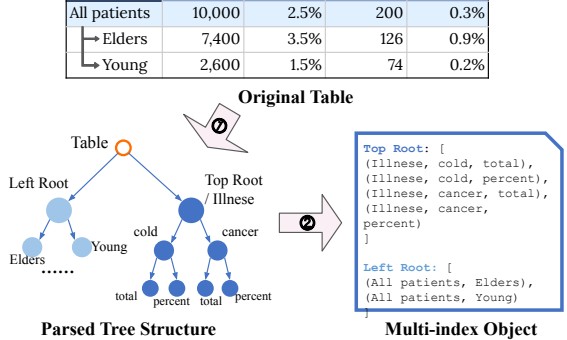

(c) A hierarchical matrix table parsed into a multi-index object: step (1) transforms table headers into a tree representation, and step (2) creates a Pandas multi-index object.

Figure 2: Examples of diversely structured tables.

**AIT-QA** (Katsis et al., 2022) contains tables and questions that particularly focus on the airline industry domain. It also contains tables with complex structures such as hierarchical headers or bi-dimensional matrix shapes. Both the complex structure and domain-specific terminology are challenging for TableQA tasks (Zhu et al., 2021; Katsis et al., 2022). AIT-QA contains 515 questions that are distributed across 116 tables. We use all 515 questions as the test set for evaluation purposes.

**Spider** (Yu et al., 2018) is a multi-domain text-to-SQL benchmark with tables in a database format. It contains 200 databases across 138 different domains. All "databases" in Spider consist of multiple relational tables connected by primary / foreign keys, which poses a unique challenge for methods – the multi-table structure. Evaluations on the hidden test set are supported by its online benchmark which only accepts SQL programs. We instead evaluate on the validation set containing 1,034 questions.

## 3 Method

We first describe our approach for representing variously structured tables with a unified multi-index format (§3.1). We then introduce our method to answer TableQA questions by generating Python programs (§3.2) from a code LLM. Lastly, we detail the use of custom helper API functions (§3.3) that are learned to be used by the LLM in context (§3.4). Figure 3 gives an overview of our method.

### 3.1 Unifying Representation of Diverse Table Structures with Multi-Index Parsing

To represent diverse table structures in a unified way that is naturally compatible with Python programs, we propose to transform any table with varying structures, denoted as $t$, into a Pandas multi-

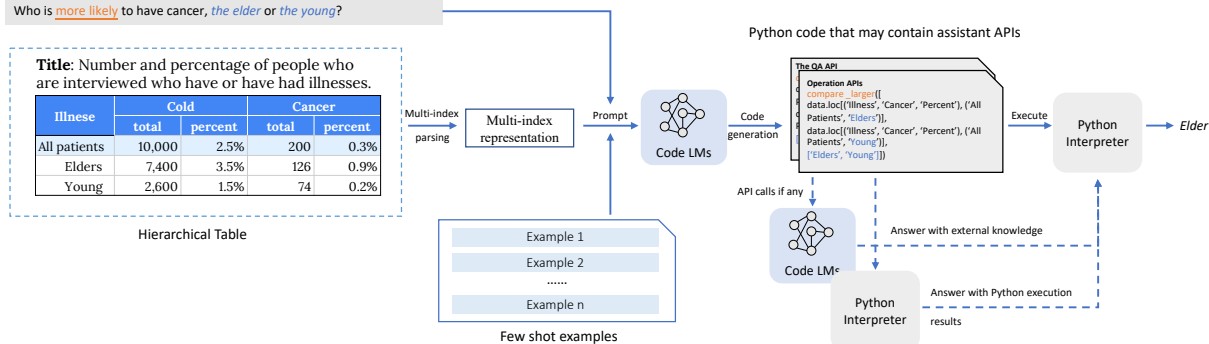

Figure 3: Our framework consists of three stages: (1) *Multi-index parsing* accepts one or more tables with diverse table structures, and transforms them into a single hierarchical table, represented using a Pandas multi-index representation $m$ (a process described in §3.1). (2) We then provide a natural language question, in-context exemplar annotations, and multi-index representation $m$ as a prompt for pretrained code models like CODEX or STARCODER for Python code generation, as described in §3.2 and §3.3. The prompt specifies examples of using our QA API and operation APIs. (3) Finally, we execute the Python code, which could contain QA or operation APIs, on the Pandas data frame, outputting the final results. Our API definitions are given in Figure 5 and Figure 4.

index representation,[1] $m$. We denote this process as *multi-index parsing*, which first converts a raw table $t$ to a bi-dimensional tree structure $b = (T, L, V)$, then parses it into a multi-index representation $m$. The two steps are illustrated in Figure 2c.

First, we first transform a table $t$ of any structure into a bi-dimensional tree representation $b = (T, L, V)$ as defined by Wang et al. (2021b). It yields two coordinate trees: (i) the left tree $L$ by indexing the values $V$ from the left header, and (ii) the top tree $T$ by indexing data cells $V$ from the top header. Specifically in the example Figure 2c, $T$ represents the deep blue area (i.e., two rows at the top); $L$ represents the first column to the left; and $V$ covers all the data cells from $10,000$ from the top-left to $0.2\%$ to the bottom right.

Next, we traverse the left and top trees and extract header path tuples from both header trees to define the row indexes and column indexes in the multi-index representation $m$, which can be used to reconstruct the original table using Pandas semantics. Specifically, we use a preorder traversal (as illustrated in Algorithm 1) on both the top tree $T$ and the left tree $L$, and enumerate all paths starting from the root node to each leaf header node. For every such root-to-leaf path, we aggregate all cell strings along the path into a tuple. For each header tree, we collect all such tuples and correspondingly define the row/column index list in the multi-index representation $m$. For example, for the top header in Figure 2c, `Illness → Cold → total` is one

---

[1] https://pandas.pydata.org/docs/reference/api/pandas.MultiIndex.html

---

**Algorithm 1** Preorder tree traversal for multi-indexing.

---

$result \leftarrow list()$
$tempList \leftarrow list()$
**procedure** PREORDER($p$: coordinate tree node, $result$, $tup$)
    **if** $p$ is not tree root **then**
        $tempList \leftarrow tempList + (p.value, )$
    **end if**
    **if** $p.index$ is None **then**
        $results \leftarrow results + tempList$
    **end if**
    **for** $child$ in $p.children$ **do** PRE-ORDER($child, result, tempList$)
    **end for**
    **return** $result$
**end procedure**

---

possible top header path, then we add (`Illness`, `Cold`, `total`) to the column index list in $m$.

Note that this *multi-index parsing* process can be readily applied to tables of diverse structures, with a unified representation efficiently preserving all structural information. In the special case of simple-structured tables such as flat relational tables, both the top and left headers are represented by single-layer trees and undergo a similar parsing process as hierarchical structures. When dealing with questions that involve multiple tables, we convert all tables into multi-index objects, feed them to the model, and let the model choose which parts it will use to answer the question. For more detailed

illustrations of this parsing process for tables of different structures, please refer to §D.

## 3.2 Generating Python Programs

With table environments represented as multi-index representation $m$, we generate Python programs $p$ to execute on $m$ to yield the answer prediction $a'$. Our method is general and not tied to any particular code generation model, as long as the model has sufficient natural language understanding and code generation capability. As such, we validate our approach using two code language models (code LMs) for Python program generation: the proprietary CODEX model (Chen et al., 2021) and the open-source STARCODER model (Li et al., 2023).

CODEX is a code LM based on GPT-3 (Chen et al., 2021). We access it through OpenAI API calls.[2] We experiment with the strongest version DAVINCI-002 with $175B$ parameters. STAR-CODER is an open-source $15B$ code LM, which at the time of this writing achieved top performance among open-source models on many code generation benchmarks such as HumanEval (Li et al., 2023). Both models are trained on a mixture of natural language and code sequences. We generate code from both models using few-shot prompting (Brown et al., 2020), with prompts specifying examples of the mapping from natural language questions and tables to Python Pandas code. More details follow in §3.4.

## 3.3 Incorporating Assistant APIs

We describe two types of API functions, Question Answering (QA) APIs and Operation APIs, to expand the ability of code LMs to incorporate external knowledge and extra table operations.

**Operation APIs** Operation APIs are designed to enhance the functionality of LM-generated programs. Since we use Python as the query language and represent tables in Pandas, we are able to incorporate simple Python APIs into the solution. For demonstration, we employ two simple comparison APIs in this paper. We emphasize that our framework is not tied to any specific operation API, and therefore offers flexibility in designing customized APIs for different datasets or question types.

The design of the comparison APIs is inspired by the empirical observation that CODEX is better at handling numerical questions, such as "what is

---

²One can call the CODEX model via API without knowing the internal process or model parameters.

---

```python
def compare_larger(values: list[float], args: list[str]) -> str:
    """Return the argument associated with the larger value."""
    return args[values.index(max(values))]
```

```python
def compare_smaller(values: list[float], args: list[str]) -> str:
    """Return the argument associated with the smaller value."""
    return args[values.index(min(values))]
```

Figure 4: Illustration of the two comparison operation APIs.

the age of Person A?", than qualitative questions like "which is higher, A or B?". We design two comparison APIs, as shown in Figure 4, to address this issue. Each API takes a list of values and a list of indices and returns the largest or the smallest index according to the values. An example showing how the comparison APIs are used is shown in Figure 1. We find in §5 that the additional operations enabled by these APIs substantially improve performance.

**QA API** Sometimes TableQA questions require external knowledge which is not included in the given table. The QA API, proposed by Cheng et al. (2023), is designed to access external knowledge when solving table QA tasks, by making calls to an LLM.

```python
def table_qa(table, question: str, values: list[float], args: list[str]) -> str:
    """Given a table, write code to answer the question"""
    ......
    mapped_qa_column = question_answering([f"Is {i} a city in Canada?"
for i in table['cities'], values, args)
    selected_columns = mapped_qa_column[mapped_qa_column==True]
    ......

def question_answering(question: list[str]):
    """Return the model answer."""
    Q = [f"Answer question with True or False. {q}" for q in question]
    A = [call_generative_model_api(q) for q in Q]
    return A
```

Figure 5: Illustration of the question answering API.

As illustrated in Figure 5, during the table question answering stage, we prompt the LLM to write questions itself. These generated questions will direct the code parsing program to call the QA API function. The QA API takes in this question and a set of table indices as parameters. For each value in the questioned column, we insert the value into the generated question and prompt the model again to give an answer to this question. Then, we select the rows with `true` values in the returned column. We include a real case from HiTab (Cheng et al., 2022) in Figure 11 which requires geographical knowledge to accurately answer the question.

### 3.4 Learning APIs In Context

Building upon the in-context learning ability of large language models (Brown et al., 2020), we use examples of API usage as a few-shot context for LM generation of programs. Our LM prompt is designed to be a mixture of instructions and solution examples. The instructions specify API usage guidelines and scenarios in which it should be used. The solution examples incorporate a balanced mix of examples, some of which utilize the assistant API to perform tasks, while others tackle the question with Pandas semantics directly. With this diversity of input examples, we aim to encourage the usage of the assistant APIs in the code generated by the LMs, while avoiding over-reliance on some APIs. Refer to §B for more complete prompt templates we used for different datasets.

## 4 Experiments

In this section, we first introduce the experiment (§4.1) and evaluation settings (§4.2), then present our results and analysis (§4.3). We also test the robustness of our method with different code models (§4.4) and intermediate query language forms (§4.5).

### 4.1 Implementation Details

We use the STARCODER 16$B$ and code-davinci-002 (CODEX) models to generate Python programs. We set the sampling temperature to $0.7$ for both models. For every question, we generate five model predictions and select the candidate with the highest log probability. We include five shots in the prompt for WikiTQ, HiTab, and Spider datasets, and eight shots for the AIT dataset.

### 4.2 Evaluation Metrics

We evaluate performance using the *Execution Accuracy* (EA) metric following Pourreza and Rafiei (2023) and Cheng et al. (2023). EA measures the percentage of questions for which our method produces programs that yield correct answers. We did not include exact matching (EM) as the measure to determine if the predicted query is equivalent to the gold query, since it can lead to false negative evaluations when the semantic parser generates novel syntax structures (Pourreza and Rafiei, 2023; Yu et al., 2018). Moreover, as Python is our main query and API function language and most datasets do not provide golden queries in Python, EM is not a practical measure in our case.

### 4.3 Experiment Results

Table 2 shows our main experiment results across the datasets described in §2.2, encompassing varied table structures, using the CODEX (code-davinci-002) code LM. We compare our full method (**Ours**) with an ablated version (**CODEX**) that only uses few-shot prompting of the code LM, without including our multi-index structure and API functions (see §5.2 for details and more ablations). We also give the performance of state-of-the-art baselines for each dataset from past work.

| Dataset | HiTab | Spider | AIT-QA | WikiTQ |
|---|---|---|---|---|
| Baseline | MAPO 40.7 | DIN-SQL 61.5 | RCI 51.8 | BINDER **54.8** |
| Codex | 59.6 | 61.2 | 77.8 | 41.7 |
| **w/ API (Ours)** | **69.3** | **63.8** | **78.0** | 42.4 |

Table 2: Execution accuracy on four TableQA dataset with CODEX generated programs, in comparison to state-of-the-art baselines: the MAPO (Liang et al., 2018) baseline from HiTab (Cheng et al., 2022), DIN-SQL (Pourreza and Rafiei, 2023) for Spider, RCI (Glass et al., 2021) for AIT-QA, and BINDER (Cheng et al., 2023).[3]

Our approach, which uses a unified representation for tables and question semantics, typically improves over past work despite their use of representations tailored to individual datasets. In particular, our approach obtains an improvement of 28.6% over the past state-of-the-art training-based method (Liang et al., 2018), and a 9.7% improvement on plain few-shot prompted Codex models for the HiTab dataset.

One exception is the WikiTQ dataset, where our method underperforms the BINDER approach (Cheng et al., 2023). We attribute this to the superior alignment of BINDER's API with the question characteristics of the WikiTQ dataset, which includes many questions that require external knowledge.

Meanwhile, we find strong performance from using Python as a QA representation of the code LM alone: in particular, on the HiTab dataset, we see that the Codex model already achieves an 18.9% improvement compared to previous training-based methods (Liang et al., 2018).

---

[3]For a fair comparison with our approach, we use 5-shot prompting for BINDER rather than the 12-shot experiments in Binder et al. (2022), which decreases performance compared to the results in their paper.

## 4.4 Robustness to Code Generation Model

We next analyze method performance when using the recently released open-source code generation model STARCODER-16B (Li et al., 2023). The findings are compiled and presented in Table 3.

While the base model performance with STAR-CODER is lower than CODEX, we still observe an improvement from our generated API across all settings, with a substantial improvement on the AIT-QA dataset, presumably because a large portion of the questions in AIT-QA can be more easily solved using our API functions. This indicates that our approach is compatible with a range of code generation LMs, and performance may scale with future improvements in open-source code LMs.

| Dataset | HiTab | AIT-QA | WikiTQ |
|---|---|---|---|
| STARCODER | 29.0 | 4.0 | 22.4 |
| w/ API | 30.1 | 26.1 | 23.2 |

Table 3: Execution accuracy on four TableQA datasets with STARCODER generated programs.

## 4.5 Robustness to Intermediate Forms

One feature of our Python and Pandas-based method for TableQA is that it is modular, and can use existing API methods from other Python libraries when appropriate. In this section, we perform a case study where we use the SQL querying API of the pandasql extension library for Pandas.[4] We experiment with the WikiTQ dataset since all tables within are relational types, the operations on which can be readily covered by the SQL grammar.

Similarly to the in-context learning configuration introduced in §3.2, we adapt our input to prompt the model to generate SQL queries. An example prompt is shown in Figure 6.

The generated SQL queries are executed on Pandas data frames and return Python objects, hence, we can still retain our same API functions and multi-index data frame representation. Please find more details about the prompt and code in §C.

The experimental results with and without the SQL API are presented in Table 4. We see that SQL outperforms Python on the WikiTQ dataset, which we suspect is because this relational dataset affords short SQL queries, which are easier to generate than long Python functions for code LMs trained on sufficient amounts of SQL. The primary

[4]https://pypi.org/project/pandasql/

*WikiTQ SQL Prompt:*
You are a helpful assistant in writing code to answer questions. Write sql query behind "SQL:" to answer the given question correctly.
All column types are set to str by default. Please first convert needed columns to the correct type if needed.
Please make sure all outputs are dataframes. Refer to these exaples:

[FEW SHOTS]

# Parse the question into SQL based on the given table below.
CREATE TABLE data(
     Year text,
  ……
     Avg. Attendance text)
[3 EXAMPLE ROWS]
# Q: what was the last year where this team was a part of the usl a-league?

Figure 6: An example prompt for a model to generate SQL queries on the WikiTQ dataset.

| Code Model | Query Language | | | |
|---|---|---|---|---|
| | Python | w/ API | SQL | w/ API |
| STARCODER | 15.6 | 15.6 | 22.4 | 23.2 |
| CODEX | 41.7 | 42.4 | 45.4 | 46.7 |

Table 4: Execution accuracy on WikiTQ when generating Python and SQL programs.

reason for the need to generate very long Python functions is to accommodate different data input types. Nevertheless, we see improvements from our API functions when applied on top of either query language in nearly all settings.

## 5 Ablation Studies

We conduct ablation analyses on the HiTab dataset to examine the improvements brought by assistant API functions (§5.1) and multi-index parsing (§5.2). We refer to multi-index parsing as *w/ MI*, and denote the opposing table flattening approach as *w/o MI* in the result tables.

| API Functions | | Multi-Index | |
|---|---|---|---|
| QA | Operation | w/o MI | w/ MI |
| ✗ | ✗ | 59.6 | 56.0 |
| ✓ | ✗ | 57.4 | 58.7 |
| ✓ | ✓ | 59.1 | 69.4 |

Table 5: Comparing CODEX execution accuracy on HiTab by removing *multi-index parsing* or helper API functions. MI stands for **M**ulti-**I**ndex parsing. For experiments without MI, we employ the flattening approach (Figure 7).

## 5.1 API Functions

We first study the improvement brought by API functions of different types. As described in §3.3, we provide the model with (i) the QA API, and (ii)

| Illness | Cold | | Cancer | |
|---|---|---|---|---|
| | total | percent | total | percent |
| All patients | 10,000 | 2.5% | 200 | 0.3% |
| Elders | 7,400 | 3.5% | 126 | 0.9% |
| Young | 2,600 | 1.5% | 74 | 0.2% |

**Hierarchical table**

*Flattening* →

| Illness | Cold_total | Cold_percent | Cancer_total | Cancer_percent |
|---|---|---|---|---|
| All patients | 10,000 | 2.5% | 200 | 0.3% |
| Elders | 7,400 | 3.5% | 126 | 0.9% |
| Young | 2,600 | 1.5% | 74 | 0.2% |

**Flattened Pandas dataframe**

Figure 7: Representing a structured table by flattening hierarchical headers.

operation APIs. In comparison to the base setting in which neither type of API is utilized, we add one type of API at a time to examine their gains.

As shown in Table 5, both types of APIs are helpful. Comparing the two API types, the operation API brings more marginal improvement than that of the QA API. In particular, adding the QA API increases performance by 2.7%, while further adding the operation APIs improves by another 10.7%, compared to the full system without APIs.

### 5.2 Multi-Index Transformation

Next, we study the contribution of adopting multi-index as a hierarchical table representation.

In comparison to our multi-index parsing baseline, we experiment with an alternative table header flattening approach that has been adopted for table processing in many works (Parikh et al., 2020; Liu et al., 2022; Wang et al., 2022). More specifically, we flatten the top header hierarchy into a single-row relational header by concatenating all cell strings from the root to each leaf header node. Note that this process ignores the header information in dimensions other than the top and treats them as normal table cells. This flattened representation is then integrated into our framework using Python queries, code models, and API calls to facilitate the ablation study. An illustration of this flattening process is shown in Figure 7.

As shown in Table 5, the execution accuracy decreases by 3.6% with the multi-index approach compared to the table flattening approach when no API is included. However, as we involve QA and operation APIs, the multi-index approach demonstrates a 10.3% improvement over that without multi-indexing. This indicates that the proposed multi-index representation is essential in preserving the rich structural information of hierarchical tables, and can be leveraged by our API operations.

## 6 Related Work

**QA over Varied Table Structures** Existing methods for table QA face many challenges both within specific table structures and across table types. SQL queries perform well in solving questions with relational tables (Pasupat and Liang, 2015; Yu et al., 2018), but their grammar is often unable to cover the entire spectrum of questions (Guo et al., 2019). Domain-specific logical forms are often favored when handling more complicated table structures (Wang et al., 2021a; Katsis et al., 2022), queries on text (Gupta et al., 2019), and grounded environments such as visual question answering (Andreas et al., 2016a,b; Johnson et al., 2017). However, these specifically designed logical forms do not support the operations needed outside their intended domain (Cheng et al., 2022). In contrast to previous methods which focus on adapting their solutions to certain table structures, we transform table structures using the multi-index format and utilize the natural flexibility within the Python language. Furthermore, we also introduce the usage of custom API functions as a modular approach to extend the framework.

**Program generation and assistant functions** Several works have leveraged program generation to answer questions or carry out planning tasks (Singh et al., 2022; Liang et al., 2022) within other environments such those involving visual modalities (Subramanian et al., 2023; Surís et al., 2023; Gupta and Kembhavi, 2023). Past work has also proposed connecting natural and programming languages by adding textual QA abilities to programs (Cheng et al., 2023) or adding programmatic operations to NL (Gao et al., 2022; Chen et al., 2022). In comparison, we focus more on extending the limited built-in operations in programming languages (e.g., Python), and incorporate helper API functions to enable broader capabilities for TableQA tasks. This is similar to the goal of extending functions into a given programming language (Herzig et al., 2020; Guo et al., 2019), however, our APIs also benefit from being easily implemented as Python functions themselves.

## 7 Conclusion

We introduced a framework for solving general TableQA tasks. This framework is built upon three core components: a unifying multi-index representation, Python programs as the query language, and code generation by prompting a large pretrained code model. Using CODEX as the code model, we demonstrate improvements over the state-of-the-art performance for multiple datasets, each with its unique table structures and challenges. Through ablations, we show that our proposed multi-index and API functions are both critical to the success of the framework, with the largest improvements on datasets involving hierarchical tables. We also observe improvements in performance for an open-source model, STARCODER, demonstrating the effectiveness of our approach with different code models.

## Limitations

We note that though our general framework consistently improves performance across datasets, we identify through ablations components in our framework that, individually applied, adversely affect performance on particular datasets. However, these same components are able to improve performance in combination with other features of the framework, as well as across other datasets. Thus, we consider these capabilities to be a step towards solving general TableQA tasks.

One limitation of our current framework is its manual definition of API functions, which may not extend to other datasets with disparate table structures, or involve richer grounded settings. Thus, a future direction for our work is to automate the API generation for different QA datasets and questions. We envision an interactive framework that, given feedback on its performance on a dataset, constructs new API functions to solve question types with lower performance.

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

## A  Few-shot Results

We further conduct experiments on WikiTableQuestions and HiTab datasets to study the influence of different shots in the prompt. Results are shown in Table 6.

| Dataset | 5 shots | 12 shots |
|---------|---------|----------|
| HITAB   | 69.3    | 70.0     |
| WIKITQ  | 42.4    | 42.8     |

Table 6: Execution accuracy on HITAB and WIKITQ with different number of prompts.

## B  Prompt Templates

Below we show the templates that we used to prompt code generation models to give answers.

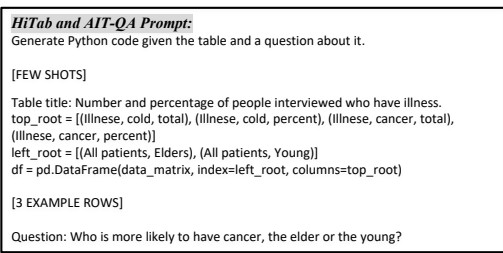

Figure 8: An prompt example for hierarchical table question answering problems.

## C  Incorporating SQL into Python Pandas

The prompt template we use for SQL querying in shown in Figure 9a and Figure 9b. The generated SQL query is wrapped in pandasql[5] queries and executed using Python SQLite tools. The return value for these queries is pandas data frames, upon which our self-designed APIs will take effect. This way of combining SQL and Python makes it possible to directly apply our Python API functions to SQL-returned results.

## D  Examples for Multi-Index Parsing

See examples of multi-index parsing in both hierarchical table and flat tables in Figure 10.

---

[5] https://pypi.org/project/pandasql/

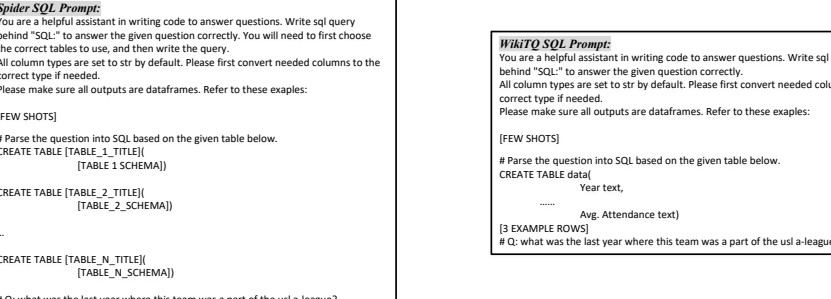

(a) An prompt example for the Spider dataset.   (b) An prompt example for the WikiTQ dataset.

Figure 9

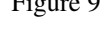

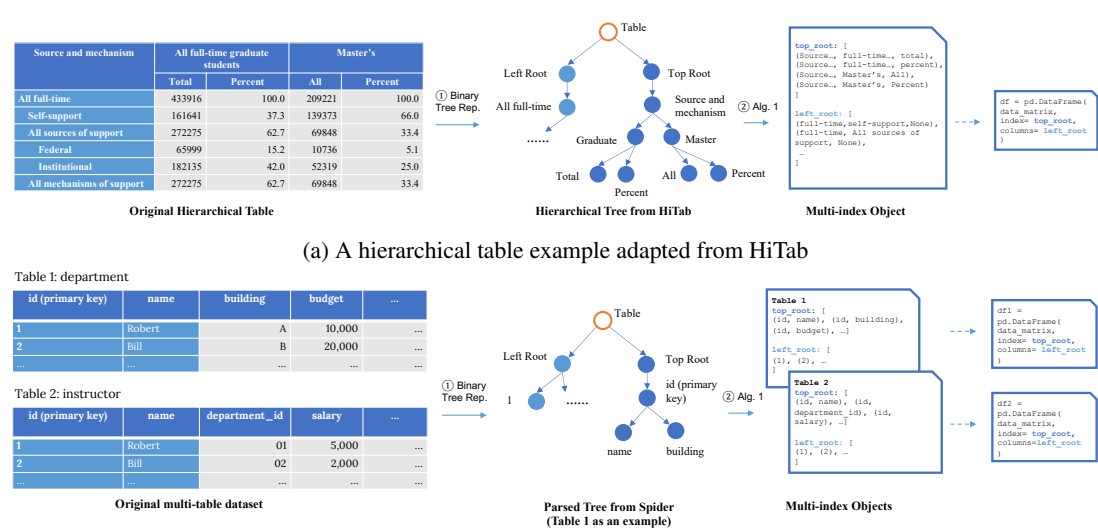

(a) A hierarchical table example adapted from HiTab

(b) A multi-table example adapted from Spider

Figure 10: Illustration of *multi-index parsing* with two datasets. We first transform the original table(s) into parsed hierarchical tree(s), similar to HiTab (Cheng et al., 2022). Then we parse the hierarchical tree(s) using depth-first traversal to form the multi-index lists. In Figure 10a, the top root is Source and mechanism, whose children define different kinds of source and mechanism of the participants, while the left root is All full-time, whose children consist of different kinds of full-time jobs. With the final output of the multi-index representation, all cells can be uniquely identified, and thus can reconstruct the table using a simple Pandas code. We also present an example of flat tables in Figure 10b, demonstrating the general applicability of our method.

| Province | Cold | | Cancer | |
|---|---|---|---|---|
| | total | percent | total | percent |
| Canada | 10,000 | 2.5% | 200 | 0.3% |
| Newfoundland and Labrador | 7,400 | 3.5% | 126 | 0.9% |
| Prince Edward Island | 2,600 | 1.5% | 74 | 0.2% |
| Winnipeg | 1,600 | 5.5% | 92 | 0.4% |
| Rest of Manitoba | 2,600 | 2.5% | 14 | 0.2% |
| ... | ... | ... | ... | ... |

Figure 11: The table is the distribution of new immigrants in different regions/provinces of Canada adapted from HiTab. One question related to this table is "how much percentage point has Manitoba rose when it comes to immigrants intended landing destination". However, the information that Winnipeg is a city of Manitoba is not included in the table itself. In such case, no method we tested could give the correct answer, which should contain a numerator that adds up "Winnipeg" and "Rest of Manitoba". In contrast, by calling QA API `qa("Manitoba?", left root, 1)` to Codex, which returns an answer `yes` to both "Winnipeg" and "Rest of Manitoba", code models could return the accurate answer to this question.