# OpenReview forum: "API-Assisted Code Generation for Question Answering on Varied Table Structures"
_EMNLP/2023/Conference — EMNLP 2023 Main_

### Official Review · Reviewer_87ms · 2023-08-01

**Typos Grammar Style And Presentation Improvements:** 1. The claim of supporting arbitraril…
**Soundness:** 3

**Excitement:**

3: Ambivalent: It has merits (e.g., it reports state-of-the-art results, the idea is nice), but there are key weaknesses (e.g., it describes incremental work), and it can significantly benefit from another round of revision. However, I won't object to accepting it if my co-reviewers champion it.

**Paper Topic And Main Contributions:**

This paper targets question-answering on structured tables, TableQA for short. Owing to the diverse types of tables, such as relational tables, hierarchical tables, and databases, previous work had proposed various approaches and used different logical forms for a specific kind of table. This paper instead proposes a unified framework for TableQA. This unification occurs in 3 levels. First, various types of tables are represented as a multi-index pandas DataFrame. Second, Python serves as a generic intermediate representation of natural language. Third, powerful code generation models are leveraged to generate Python code in a few-shot manner. In addition, this framework allows customs API functions to extend the Pandas functionality and to incorporate external knowledge. Experimental results on few-shot HiTab, Spider, and WikiTQ demonstrate the performance of the framework to some extent.

**Questions For The Authors:**

1. Please elaborate on the considerations of baseline approaches.
2. Please clarify the gain from multi-index transformation and API. It would be better if we can have the same ablation on Spider, AIT-QA, and WikiTQ (optional).

**Reasons To Accept:**

1. A simple and generic framework for TableQA. This framework is especially helpful for tables that cannot be queried by SQL and is also good at leveraging the strong Python code generation capability of advanced LLM and code generation models.

2. The paper is well-written and easy to follow.

**Reasons To Reject:**

My primary concern about this paper is the experiment part.

1. Baseline approaches selection. This paper compares 4 baseline approaches in the experiment, but it does not explain the reasons for choosing them or the baseline settings. For example, DIN-SQL using Codex achieves far better performance (69.9%) than the one reported in Table 2 (61.2%). Also, the proposed approach can align with BINDER in WikiTQ, i.e., using 12-shots, but this paper chose to use 5-shot and did not show the performance by the number of few-shot examples. The reasons for such settings are essential for readers to understand the performance and superiority of the proposed approach.

2. The performance gain of using multi-index transformation and API is not impressive except for HiTab. Specifically, compared with Codex, w/ MI and API only provides 2.6%, 0.2%, and 0.7% absolute improvement on Spider, AIT-QA, and WikiTQ, respectively. Also, we can observe from Table 5 that only using MI itself even hurts the performance.

**Reproducibility:**

4: Could mostly reproduce the results, but there may be some variation because of sample variance or minor variations in their interpretation of the protocol or method.

**Reviewer Confidence:**

5: Positive that my evaluation is correct. I read the paper very carefully and I am very familiar with related work.

---

> ### Author Rebuttal · Authors · 2023-08-29
>
> We thank the reviewer 87ms for thoughtful feedback. Your positive comments on our approach to deal with question answering over hierarchical tables are truly encouraging. For your concerns and questions about our work, please find responses below.
>
> We would like to emphasize that one contribution of our work is the use of Pandas DataFrames as a unified representation for tables of varied structures, in conjunction with prompting LLMs. We will clarify this framing in the paper, which we believe should help to address two of your points: (1) For your point that SQL is a strong querying language for multi-table datasets – we absolutely agree! Our Pandas representation is also compatible with SQL; as our experiments in Sec 4.5 show – finding improvements of 6.8% with StarCoder and 3.7% with Codex on WikiTQ by using SQL instead of Python as querying language. We will clarify our positioning, and make it clear that we do not claim that Python is the optimal query language, only that we advocate for Pandas as a unified representation. (2) We show that LLMs, when combined with our Pandas representation, achieve strong, often state-of-the-art performance across datasets. While our additional multi-index and API contributions do produce the largest improvements on HiTab, our Pandas+LLM baseline (without multi-index and APIs), referred to as “Codex” in the results tables, is also a contribution of our work. We will make these contributions clearer in the paper.
>
> We will also refine our statements about arbitrarily-structured tables – while our representation handles both the range of table structures seen in our 4 datasets in the paper, and can also handle other sorts of tables (see our response to rUJ6 about multi-section tables), we agree that “arbitrary” was strong wording, and are willing to change to language like “unified representation for varied table structures”, or other suggestions you might have, in future versions of the paper.
>
> - **Q1 & R1. Considerations of baseline selection**
>
> Thanks for pointing this out. As mentioned in the caption of Table 2, for each dataset, we selected baselines based on the state-of-the-art methods from the literature at the time of submission. For the case of DIN-SQL – their paper presents both few-shot performance results, and results with the incorporation of an additional self-correction generation mechanism, which prompts one model several times iteratively to finally complete one generation pass. As our method focuses on table representation and prompt design, for fair comparison we compare to their few-shot performance. Incorporating a self-correction mechanism like DIN-SQL’s on top of our table representation and prompt design strategy is a promising direction for future work.
>
> Additionally, we present few-shot performance on Codex, especially since datasets such as HiTab and AiT-QA remain relatively underexplored. We recognize that there are other approaches proposed to improve the few-shot performance, however, they are concurrent works during our submission period.
>
> - **R1. few-shot results with different shots not shown**
>
> We provide further experiment results comparing 5-shot and 12-shot prompt as follows:
> | Dataset | 5 shots | 12 shots |
> |---|---|---|
> | HiTab |69.3 | 70.0 |
> | WikiTQ |42.4 | 42.8 |
>
> From the results, we can see that with adding the number of shots in the prompt, the performance didn’t improve a lot. Also, 5-shots is more compatible with open-source code generation models which have a limited context window.
>
> - **R2. Effectiveness of Multi-Indexing (MI)**
>
> First, multi-indexing is easier to use than implementing four different data processing and table querying pipelines for different datasets. In addition, MI brings better performance on all types of structures. This improvement appears the most significant on HiTab (which contain hierarchical tables), but doesn't impair its efficacy on any other datasets. Intuitively, compatible data processing and table querying schemes provide the best result, whereas Table 5 suggests that it is best to use MI together with our Python-based QA schema. Conversely, simply unifying data processing with little consideration of the query step (only using MI) may lead to degraded performance.
>
> - **Q2. Gain from MI and API on Spider, AIT-QA and WikiTQ**
>
> For Spider and WikiTQ (relational) dataset, using MI is basically the same as using the normal representation of a table. Therefore, the performance improvement on Spider and WikiTQ mainly comes from the API. We only performed ablation studies on HiTab since it can properly reveal the benefits of MI and API. We will add ablation results on other datasets in the revised paper version.

---

### Official Review · Reviewer_VuHH · 2023-08-02

**Soundness:** 4

**Excitement:**

4: Strong: This paper deepens the understanding of some phenomenon or lowers the barriers to an existing research direction.

**Missing References:**

No missing references

**Paper Topic And Main Contributions:**

The authors propose a novel approach to solve general table QA tasks, rather than limiting it to some specific table formats. They introduce a framework to solve this problem, which uses a unifying multi-index representation, assistant API functions, and Python program generation by prompting a large pre-trained code model.


**Questions For The Authors:**

Now you just provide a small number of API functions, and you use a few-shot to prompt. But if we want to extend this method's capability, it is likely to add more API functions, which need more few-shot examples to prompt.  Have you thought of this question and how do you think to solve this problem?

Although your method achieves start-of-the-art performance on HiTAB, Spider, and AIT-QA datasets, its performance on the WikiTQ dataset is not so good and you attribute this to question characteristics of the WikiTQ dataset, which includes many questions that require external knowledge. Have you thought about how to improve its performance, such as using some APIs that can be queried on the Web? And more, have you thought about how to make it answer questions that require specialized domain knowledge?

Now you define some operation APIs and QA APIs in advance. Do you think it is a good idea to train a model which can make necessary APIs automatically because there may be cases that it needs a new API that you haven't defined in advance?

**Reasons To Accept:**

The paper presents a novel and innovative approach to address the general table QA tasks. It is quite novel and impressive to convert tables into multi-index representation, while in traditional methods, tables are serialized to text sequences. Moreover, the paper provides a thorough experimental evaluation, doing ablation studies and other comprehensive evaluations to make the conclusion more convincing and reliable.
And the presented approach achieves state-of-the-art performance on benchmark datasets, surpassing existing methods. Furthermore, the paper is well-written and effectively communicates the proposed approach, making it accessible and understandable to the NLP community.


**Reasons To Reject:**

The authors should include a more extensive comparison with relevant baselines to showcase the superiority of their method convincingly. Now you just display one baseline for each dataset to compare. The paper may benefit from a more comprehensive review of related literature, including recent and relevant works in table QA tasks.


**Reproducibility:**

4: Could mostly reproduce the results, but there may be some variation because of sample variance or minor variations in their interpretation of the protocol or method.

**Reviewer Confidence:**

4: Quite sure. I tried to check the important points carefully. It's unlikely, though conceivable, that I missed something that should affect my ratings.

**Typos Grammar Style And Presentation Improvements:**

I didn't find any typographical or grammatical errors and I think this paper is well-organized and clearly written.

---

> ### Author Rebuttal · Authors · 2023-08-29
>
> Thank you for your review and for your thoughtful feedback! We are glad for your positive comments on the novelty of our approach to addressing general table QA tasks via the multi-index approach, and your comments on the thoroughness of our experiments and ablation studies.
>
> - **R1. Extensive Comparison with Baselines**
>
> We agree that sufficient baseline comparisons help solidify our findings. We would like to clarify that the baselines listed in Table 2 are not the only baselines we considered, but the *best* baseline methods according to individual datasets, as described in the caption of Table 2. We include them as representatives of all the competitive methods evaluated on each dataset. The fact that our method outperforms these previous SoTA baselines sufficiently shows that our method also outperforms other baseline methods.
>
> Additionally, the row “Codex” in Table 2 denotes another baseline we built to quantify the performance improvement due to the proposed multi-index structure and API functions. The Codex baseline incorporates few-shot prompting with Python code generation using flattened tables as a vanilla method to handle complex table structures (a process described in 5.2), and is also why it already outperforms some of the state-of-the-art baselines.
>
> In light of your feedback, we will add more explanations about the baselines in the experiment section and Table 2&3 in the revised paper version.
>
> - **Q1. Adapting to More APIs**
>
> In addition to extending the prompt to include examples of all newly introduced APIs, one could also add selection/retrieving modules to filter the most relevant ones, and only include an efficient and acceptable number of API examples to our framework. Nonetheless, that would demand designing a whole additional set of approaches which is currently beyond the scope of this work.
>
> Overall, we believe our paper provides a structured framework with unified table representation and API capability, with Operation and QA APIs as two working examples.
>
> - **Q2. Improving WikiTQ Performance**
>
> Thanks for pointing out possible ways to improve the performance over WikiTQ. While we refrained from over-tuning our method towards specific datasets (e.g., micro-adjusting hyperparameters like number of shots or adding customized APIs, as well as prompt engineering) to evaluate the generalizability of our method, we think the following methods could be compatible with our framework to improve the performance on certain datasets:
> 1. Additional effective APIs to query external knowledge could help. For example, using our framework, one can incorporate APIs designed specifically to search for relative knowledge from web, which can potentially improve the performance.
> 2. Add chain-of-thought when utilizing the API to better help the model understand why the API is used under this situation. Large language models can be confused of when and why to use the APIs even they are given with few-shot examples. To help them better understand the usage of the API methods, we can potentially add chain-of-thought explanation to few-shot examples.
> 3. More specialized data query APIs for domain-specific knowledge (e.g., internal database) could help.
>
> - **Q3. LLMs Create APIs Automatically**
>
> Yes, we have indeed considered and experimented in this direction, in light of other promising findings [1,2]. However, at the time of experimenting, neither GPT3.5 nor Codex could generate viable API functions in our setting, presumably due to limited capability or the broad variety of questions. This is understandable, as generating APIs is much more complicated than using them. We have also considered automatically generating APIs in chain-of-thought style, but found it requires substantial effort and is more suitable for a future research project.
>
> [1] Cai, Tianle, et al. "Large language models as tool makers." arXiv preprint arXiv:2305.17126 (2023).
>
> [2] Qian, Cheng, et al. "CREATOR: Disentangling Abstract and Concrete Reasonings of Large Language Models through Tool Creation." arXiv preprint arXiv:2305.14318 (2023).

---

### Official Review · Reviewer_rUJ6 · 2023-08-04

**Soundness:** 4

**Excitement:**

4: Strong: This paper deepens the understanding of some phenomenon or lowers the barriers to an existing research direction.

**Missing References:**

No missing reference.

**Paper Topic And Main Contributions:**

This paper proposes to generate executable programs for varied table structures, which is interesting. It transforms the tables to multi-index data frames in Panda Python library, then generated python programs are executable on them. It also introduces a set of special API functions to achieve broader coverage of queries.

**Questions For The Authors:**

If a hierarchical table has multiple sections, e.g., rows 1-2 are about gender, rows 3-10 are about different age groups, rows 11-20 are about funding types, how can this table be transformed to multi-index representation? One multi-index representation object is ok or multiple objects are needed?

**Reasons To Accept:**

1. This paper proposes a novel way to generate executable programs for varied table structures. For examples, hierarchical tables are commonly-used but largely been neglected by previous research works.
2. Multi-index representation is a novel way.
3. Experiment results are significant on several important and related datasets.
4. Good writing and easy to follow.

**Reasons To Reject:**

The experiment results on WikiTQ is not significant.

**Reproducibility:**

4: Could mostly reproduce the results, but there may be some variation because of sample variance or minor variations in their interpretation of the protocol or method.

**Reviewer Confidence:**

5: Positive that my evaluation is correct. I read the paper very carefully and I am very familiar with related work.

**Typos Grammar Style And Presentation Improvements:**

-

---

> ### Author Rebuttal · Authors · 2023-08-29
>
> Thank you for your review, and for recognizing the novelty and effectiveness of our work! We are encouraged that you find novelty in our multiindex representation as a unified hierarchical table representation to solve the important representation issue in using LLMs to solve QA tasks, in particular for complicated and diverse tables.
> - **R.1 WikiTQ Improvement**
>
> While this improvement still demonstrates the generalizability of our method, e.g., to extend to arbitrary table structures (including relational tables), we believe that one major reason for this small gain is the simplicity of the questions and tables in WikiTQ, where complex operations supported by advanced APIs are not necessary for most examples. As we note in Table 1, WikiTQ only contains single table structures from a single data source, while all other datasets where we demonstrate more significant gains contain questions over either hierarchical or multiple tables.
>
> Additionally, we would like to note that the focus of Table 1 is to demonstrate the wide applicability of our method across all datasets, with the comparison against the SOTA serving to help inform the reader. We also refrained from over-tuning our method towards specific datasets (e.g., micro-adjusting hyperparameters like number of shots or adding customized APIs, as well as prompt engineering) to evaluate the generalizability of our method.
>
> - **Q.1 Multiindex representation for multi-section tables**
>
> Yes, a multi-section hierarchical table can be represented with a single multi-index object. We can simply split the multiple sections and organize them with respective head nodes, serving as children nodes for the left root in one bi-dimensiontal tree.
>
> As per the example you give, the rows of the original table are (we assume there is only one column “Amount” with two sub columns “total” and percent for simplicity):
> |Category |
> |---|
> |gender  —> 1st section, Gender|
> |g_1 |
> |g_2 |
> |age —> 2nd section, Age |
> |a_1 |
> |a_2 |
>  …
> |a_N|
> |funding type —> 3rd section, Funding|
> |f_1 |
> |f_2 |
>  …
> |f_M|
>
> where gender, age, and funding type are three child nodes of left_root, Category. So such a table can be transformed into a single multi-index representation using our bi-dimentional tree representation and Algorithm 1 as:
> ```
> top_root = [(Amount, total), (Amount, percent)]
> left_root =  [(Category, gender, g_1), (Category, gender, g_2), (Category, age, a_1), (Category, age, a_2), …, (Category, age, a_N), (Category, funding type, f_1), (Category, funding type, f_2), …, (Category, funding type, f_M)]
> ```
> Subsequently it can be transformed into a Python dataframe with:
> `df = pd.DataFrame(data_matrix, index= top_root, columns= left_root)`
>
> In addition, we also refer to a similar example and more details description of multi-index transformation process in Figure 10 (a) in Appendix C of our paper (more details about the example table can be found in Figure 1 in the HiTab paper [1]). The two examples show the compatibility of our method with multiple hierarchical sections included within one table. In addition, we note our proposed method is not only compatible with multiple hierarchical sections (as described above), but also multiple levels within each hierarchical section by expanding the depth of tree representation and using Algorithm 1 in our paper.
>
> [1] Zhoujun Cheng, Haoyu Dong, Zhiruo Wang, Ran Jia, Jiaqi Guo, Yan Gao, Shi Han, Jian-Guang Lou, and Dongmei Zhang. 2022. HiTab: A hierarchical table dataset for question answering and natural language generation. In Proceedings of the 60th Annual Meeting of the Association for Computational Linguistics (Volume 1: Long Papers), pages 1094–1110, Dublin, Ireland. Association for Computational Linguistics.

---

### Meta-Review · Area_Chair_nYZ5 · 2023-09-16

**Recommendation:** 4

**Metareview:**

This paper targets question-answering on structured tables, TableQA for short. Previous work has proposed various approaches and used different logical forms for a specific table due to the diverse types of tables, such as relational tables, hierarchical tables, and databases. This paper instead presents a unified framework for TableQA. This unification occurs in 3 levels:
 1. Various types of tables are represented as a multi-index pandas `DataFrame`.
 2. Python serves as a generic intermediate representation of natural language.
3. Powerful code generation models are leveraged to generate Python code in a few-shot manner.

In addition, this framework allows customs API functions to extend the Pandas functionality and incorporate external knowledge. Experimental results on few-shot HiTab, Spider, and WikiTQ demonstrate the framework's performance to some extent.

**Positive:** The reviewers have liked the simplicity and novelty of the work.

**Negatives:** More extensive comparison with the relevant baselines and elaborating the rationale behind the choice of the baselines (VuHH, 87ms): the authors have agreed. They promised to elaborate on the existing baselines and add more. Similarly, it has raised concerns about the limited choice of baselines, which seem arbitrary. Given the number of questions I see in the reviews, there is a need to improve the writing of the paper. I hope the authors revise their work for future revisions.


**Given the "soundness" scores, I support accepting this paper. I am also leaning toward suggesting it for the "main track."**

---

### Decision · Program_Chairs · 2023-10-07

**Decision:**

Accept-Main

**Comment:**

This paper targets question-answering on structured tables, TableQA for short. Previous work has proposed various approaches and used different logical forms for a specific table due to the diverse types of tables, such as relational tables, hierarchical tables, and databases. This paper instead presents a unified framework for TableQA. This unification occurs in 3 levels:
 1. Various types of tables are represented as a multi-index pandas `DataFrame`.
 2. Python serves as a generic intermediate representation of natural language.
3. Powerful code generation models are leveraged to generate Python code in a few-shot manner.

In addition, this framework allows customs API functions to extend the Pandas functionality and incorporate external knowledge. Experimental results on few-shot HiTab, Spider, and WikiTQ demonstrate the framework's performance to some extent.

**Positive:** The reviewers have liked the simplicity and novelty of the work.

**Negatives:** More extensive comparison with the relevant baselines and elaborating the rationale behind the choice of the baselines (VuHH, 87ms): the authors have agreed. They promised to elaborate on the existing baselines and add more. Similarly, it has raised concerns about the limited choice of baselines, which seem arbitrary. Given the number of questions I see in the reviews, there is a need to improve the writing of the paper. I hope the authors revise their work for future revisions.


**Given the "soundness" scores, I support accepting this paper. I am also leaning toward suggesting it for the "main track."**